# Exploring the status of brucellosis in pregnant women presented with febrile illness at different healthcare facilities of Vehari and Lodhran zones of Pakistan

Syed Muhammad Tauseef Shafqat[1,2�‡*], Mian Muhammad Awais [1�‡*], Ulas Acaroz[3,4], Masood Akhtar[1], Abdul Sammad Ali Khan Shirwany[1,5], Imran Rasool[1], Naseer Ali Shah[2]

1 One Health Research Laboratory, Department of Pathobiology, Faculty of Veterinary Sciences, Bahauddin Zakariya University, Multan, Pakistan, 2 Department of Biosciences, Faculty of Sciences, COMSATS University, Islamabad, Pakistan, 3 Department of Food Hygiene and Technology, Faculty of Veterinary Medicine, Afyon Kocatepe University, Afyonkarahisar, Türkiye, 4 Department of Food Hygiene and Technology, Faculty of Veterinary Medicine, Kyrgyz-Turkish Manas University, Bishkek, Kyrgyzstan, 5 University Institute of Public Health, Faculty of Allied Health Sciences, The University of Lahore, Lahore, Pakistan

उ These authors contributed equally to this work.
‡ These authors share the first authorship on this work.
* muhammadawais@bzu.edu.pk (MMA); tauseefshafqat@yahoo.com (SMTS)

## Abstract

### Background

Brucellosis is an important zoonotic disease which causes stillbirth and abortion in pregnant women. It remains undiagnosed due to nonspecific symptoms like febrility, a very common ailment sign. Accordingly, this study was conducted to determine the seroprevalence and risk-determinants of brucellosis among pregnant women presented with febrile condition to healthcare facilities of Vehari and Lodhran regions of southern Punjab-Pakistan.

### Methods

For this cross-sectional study, a total of 250 pregnant women with febrile condition were approached. Of these, 200 willing women were included in this study. Blood samples of participants were analyzed for sero-detection of brucellosis using serum agglutination test (SAT) and commercially available indirect-ELISA kits (sensitivity and specificity>95%) followed by detection of *Brucella* (*B.*) species using PCR. Univariate and multivariate analyses were employed to determine the risk factors associated with brucellosis.

**Data availability statement:** The authors confirm that all data supporting the findings of this study are included in the article and supplementary information.

**Funding:** This work was supported by Pakistan Agricultural Research Council under Agricultural Linkages Program (vide grant No. AS 173 (F. No. 3-141/2019 (ALP)-P&DD-PARC dated 09-10-2020). The funder had no role in study design, data collection and analysis, decision to publish, or preparation of the manuscript.

**Competing interests:** The authors have declared that no competing interest exists.

## Results

Overall seropositivity rate of brucellosis by SAT in study population was 18% whereas ELISA revealed the seroprevalence rate of 12% (n=24/200). PCR findings showed the presence of *B. abortus* and *B. melitensis* in seropositive participants. Analysis revealed that nutritional status (P=0.037, OR=0.2431), educational status (P=0.049, OR=0.2168), contact with *Brucella*-susceptible animals (P=0.009, OR=10.5142), abortion history (P=0.012, OR=8.7308), raw milk's consumption (P=0.002, OR=37.1499) and lack of disease awareness (P<0.001, OR=0.0340) were significantly associated risk factors with brucellosis. Data regarding clinical manifestations revealed the highest frequency (87.50%) of fatigue with general weakness and the lowest one of night sweats (20.83%) in seropositive women.

## Conclusions

Brucellosis is prevalent in febrile pregnant women of study area. It is suggested to devise disease control/prevention measures which may include but not limited to enhancing awareness about disease-dynamics, improving disease diagnostic facilities and immunization of susceptible animals from where disease originate.

## Author summary

Brucellosis is a significant bacterial infection with zoonotic implications. It is usually transmitted from animals to humans through direct contact with infected animals or by consuming unpasteurized dairy products. According to the WHO, it is ranked as one of the most widespread zoonoses in the world, with >0.5 million cases annually. In humans, particularly pregnant women, it causes a non-specific febrile illness, a common sign/symptom in various other reproductive disorders. Due to this reason, it is usually overlooked as a potential cause of abortion during diagnosis. Subsequently, it remains misdiagnosed and under-rated, worsening the situation in endemic regions including Pakistan. This study reports the prevalence of brucellosis in pregnant women presented with febrile illness at healthcare facilities in Vehari and Lodhran regions of Pakistan. Blood samples and epidemiological data were collected and analyzed using SAT, ELISA, and PCR. Our results revealed that seropositivity rates of brucellosis in target population were 18% (SAT) and 12% (ELISA), whereas PCR confirmed the presence of *B. abortus* and *B. melitensis* in seropositive cases. Poor nutritional status, close contact with susceptible animals, abortion history, raw milk consumption, low education and limited disease awareness of study participants were found to be significantly associated risk factors of brucellosis in target population. Study findings are anticipated to assist the policy makers to devise targeted disease control programs. They will also draw the physicians' attention towards this important pathogen, when diagnosing reproductive ailments in pregnant women, especially in those presented with febrile illness in endemic regions.

## Background

Human brucellosis, also known as Malta fever, Undulant fever, or Mediterranean fever, is a bacterial infection caused by different species of the genus *Brucella* (*B.*) [1,2]. To date, more than 12 species of *Brucella* have been documented, out of which four species including *B. melitensis, B. abortus, B. suis* and *B. canis* have been reported to infect human beings [3,4]. Apart from its zoonotic implications, it also causes huge economic losses in susceptible animals and thus exerts a negative impact on the growth of livestock industry with losses up to US $ 3.4 billion in India and US $ 600 million in Latin America annually [5,6]. The transmission of brucellosis from human to human is usually rare and human beings are considered as its dead-end hosts [7]. Most commonly, it spreads to people through direct or indirect contact with *Brucella* infected animals and/or consumption of their products. In some previous studies aerosol transmission had also been reported [8].

The risk of getting infected by *Brucella* rises many folds in humans who are involved in raising/handling the susceptible animal species including cattle, buffalo, sheep and goat due to the zoonotic transmission of *Brucella* spp. from infected animals to humans who spend most of their time with their animals [9]. The disease control in human population can only be achieved by controlling disease in animals and using appropriate preventive measures while dealing with animals [3]. According to World Health Organization (WHO), it is one of the most widespread zoonoses in the world with an incidence rate of >0.5 million cases every year [10]. However, it is speculated that the real load of infection in human population is much higher due to multiple reasons including misdiagnosis, inappropriate disease surveillance system and underreporting due to limited diagnostic facilities in underdeveloped/developing countries. Due to the reason, it has been ranked as one of the world's leading neglected zoonotic diseases in low-income countries and 2nd most contagious disease across the globe [7]. Most of the developed countries have successfully controlled brucellosis by strict implementation of control measures including vaccination of healthy animals, routine screening and culling of infected animals but it is still endemic in the Mediterranean basin, the Middle East, Latin America, the Indian subcontinent, and many African countries of north and south of the Sahara [11,12].

In infected humans, it is clinically manifested by fever, fatigue, weakness, headache, nausea, weight loss, myalgia, joint pain and night sweats [13,14]. These symptoms are difficult to distinguish from other common infections due to similar clinical signs and symptoms, like those of malaria which is also very common in lower- and lower middle-income countries. Despite substantial and prolonged morbidity, early diagnosis and treatment is very difficult, and literature revealed that 62.5% cases of brucellosis are initially misdiagnosed [15]. The outcomes of brucellosis might be more adverse in pregnant women as compared to healthy and non-pregnant women due to pregnancy related immunological and physiological stress where it may lead to premature deliveries, abortions, intrauterine fetal deaths and neonatal brucellosis and other pregnancy-related complications [16–20].

Several epidemiological studies have been conducted on brucellosis in different human populations but studies targeting the pregnant women especially with febrile conditions as a high-risk group are scarce with few observational studies carried out in Pakistan [21–23]. Moreover, undulant fever is one of the hallmark symptoms of brucellosis and as it overlaps with other common febrile conditions, it is frequently overlooked in the diagnostic process, especially in resource-constrained healthcare settings where laboratory capacity is limited. This diagnostic oversight can contribute to mismanagement of febrile illnesses during pregnancy, which significantly increases the risk of adverse maternal and fetal outcomes such as spontaneous abortion, stillbirth, and preterm labor.

Keeping in view, this study aimed to determine the prevalence and associated risk determinants of brucellosis in pregnant women presented with febrile conditions at different reproductive healthcare facilities of two agroecologically important zones, Lodhran and Vehari of southern Punjab, Pakistan. Findings of this study will advocate the need to include brucellosis in national health action plan and to attract the attention of stakeholders from decision making bodies for devising prophylactic measures and healthcare infrastructure to avoid reproductive complications in this high-risk group.

## Materials and methods

### Ethical statement

This study was approved from institutional Ethics Committee of Bahauddin Zakariya University, Multan-Pakistan (AWEC/D-004/2020) and COMSATS University, Islamabad-Pakistan (CUI-Reg/Notif-1001/22/039). Informed written/verbal consent was also obtained from all participants for using data generated from lab analysis of their blood samples for research publication in an anonymous manner.

### Study area

This study was carried out in two densely populated zones (Lodhran and Vehari) of southern Punjab, Pakistan with limited healthcare facilities (Fig 1). Lodhran (29.6869° North, 71.6673° East) and Vehari (29.9719° North, 72.4258° East) are bounded by Khanewal and Sahiwal on the northern side, Pakpattan and Bahawalnagar on the east, Multan on the west and Bahawalpur on the southern side. The study area spreads over 7,142 km$^2$ with a total population of 5.2 million. The study area has extreme climatic conditions, i.e., hot and humid in summers (mean temperature 44.5°C) while cold and dry during winters (mean temperature 25.5°C) [24].

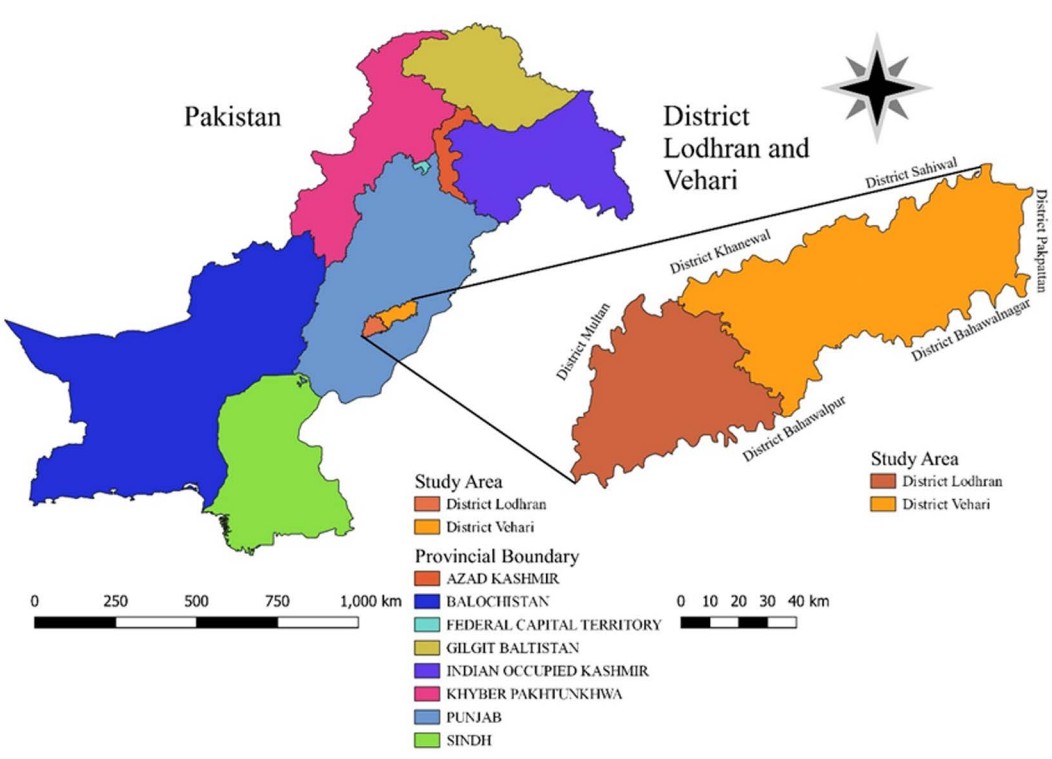

**Fig 1. Map of study area showing districts Lodhran and Vehari of Pakistan.** The map was developed using freely accessible software QGIS 3.24.2 (available at: https://qgis.org/) and base map shapefiles were obtained from the Humanitarian Data Exchange (HDX), United Nations Office for the Coordination of Humanitarian Affairs (OCHA), dataset: "Pakistan Union Council Boundaries along with other admin boundaries dataset", file: "Adminbdy Shapefile.rar" (Available at: https://data.humdata.org/dataset/pakistan-union-council-boundaries-along-with-other-admin-boundaries-dataset), licensed under Creative Commons Attribution 4.0 International (CC BY 4.0).

## Study population

This study targeted pregnant women with febrile conditions who visited gynecology outdoor patient department (OPD) of district headquarters (DHQ) hospital of district Lodhran and different maternity clinics of district Vehari from October 2021 to October 2022. Pregnant women of any age, socio-economic status and gestational age were eligible for inclusion in this study, provided they were experiencing febrile conditions. Exclusion criteria included absence of febrile illness at the time of data collection, presence of comorbid conditions with a confirmed diagnosis (e.g., malaria, dengue, typhoid fever or COVID-19) and women attending the selected healthcare facilities who were not residents of Vehari or Lodhran districts. For this analytical cross-sectional study, a total of 250 pregnant women met the inclusion criteria who were approached for blood sampling. Of whom 200 women (median age = 28 years) agreed to participate in this study ([Fig 2]).

## Sampling and collection of descriptive epidemiological data

Blood (5mL) was aseptically collected by venipuncture from each respondent using sterile syringes; 3 mL blood was immediately transferred to pre-labelled gel activated vacutainers to harvest the serum for sero-detection of brucellosis and remaining 2 mL in EDTA coated vacutainers for DNA extraction and amplification by polymerase chain reaction (PCR). To predict the influence of different risk factors on the prevalence of brucellosis in pregnant women with febrile illness, the descriptive epidemiological data including age, body mass index (BMI) category, nutritional status, education, occupation, residence, animal presence at home, animal contact, consumption of raw milk and undercooked meat, gestational age, number of normal pregnancies, history of premature birth and abortion and awareness regarding brucellosis were recorded on a pre-designed questionnaire by face-to-face interviewing of study respondents.

## Sero-detection of brucellosis by SAT and IgM-captured ELISA

All the collected sera samples were screened for seropositivity against *B. abortus* and *B. melitensis* using commercially available serum agglutination Test (SAT) reagent [Catalogue No. AFA/018 (*B. abortus*); Catalogue No. AFA/020 (*B. melitensis*); Antec Diagnostic Products, UK] as per manufacturer's instructions. All the samples were also analyzed using commercially available IgM-captured ELISA kit (Catalogue No. BA053M; Calbiotech, USA; sensitivity = 96%, specificity = 99%) as per manufacturer's protocol. Optical density was measured at 450nm wavelength using an ELISA reader (ELx800, BioTek, USA). Results were interpreted using the formulae/calculation provided by the manufacturer on instruction leaflet.

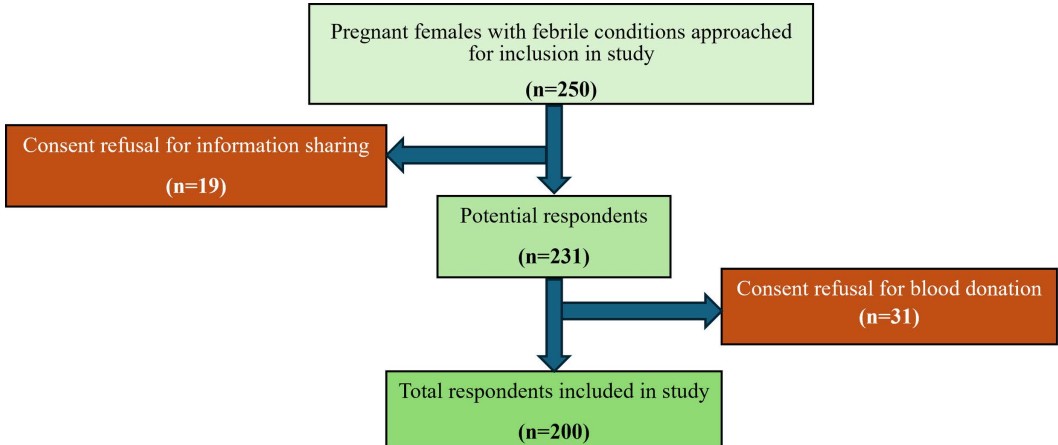

**Fig 2. Scheme of inclusion of participants in the study.**

## Polymerase chain reaction for detection of *Brucella* species in seropositive samples

The ELISA-positive samples were further analyzed by PCR using genus and species-specific primers. For this purpose, the DNA of all ELISA-positive samples (n = 24) were extracted using DNA-Extraction kit (GeneJET-Genomic DNA-Purification Kit, Thermo Fisher Scientific; Catalog No. K0721). The confirmation of *Brucellae* in samples was done by using primers targeting the *bcsp31*, *alkB* and BMEI1162 genes for *Brucella spp.*, *B. abortus* and *B. melitensis,* respectively as described previously [25].

## Statistical analysis

The data obtained from questionnaires and laboratory testing of blood samples of study respondents were entered on Microsoft Excel sheet. The Univariate analysis of risk factors was performed using R language (version 4.2.0) with RStudio (2022.02.3 + 492) as interface whereas multivariate analysis was performed by Binomial Logistic Regression Model (BLRM) using Jamovi (version 2.3.18.0). The variables with p-values less than 0.05 were considered significant.

## Results

Results showed seropositivity rate of 13.00% (n = 26/200) for *B. abortus* and 5.00% (n = 10/200) for *B. melitensis* by SAT. The co-existence of both species was also recorded in 3.50% (n = 7/200) sera samples of target population. The analysis of sera samples using IgM-captured ELISA showed an overall seroprevalence rate of 12% (n = 24/200) in study population (Table 1). Analysis of the ELISA-positive samples by genus-specific PCR, confirmed the presence of *Brucella* in all the analyzed samples (100%; n = 24/24). In species-specific PCR, *B. abortus* was detected in 70.83% (n = 17/24) of ELISA-positive samples whereas *B. melitensis* was found in 33.33% (n = 8/24) of positive samples. The co-existence of both species was also recorded in 8.33% (n = 2/24) of seropositive samples whereas in one sample (4.17%) *Brucella* species remained unidentified (Fig 3).

## Univariate analysis

Results of univariate analysis of association of host-related and socio-cultural factors with brucellosis is given in Tables 2 and 3, respectively. The highest prevalence (17.33%) was observed in pregnant women aged ≤25 years (OR=Ref.) and the lowest (7.41%) in age group of 40–50 years (OR=0.38; 95%CI OR=0.08-1.81) however the difference was statistically non-significant (p = 0.192). The association of body mass indices of target population with prevalence of brucellosis was also non-significant (p = 0.189) however underweight respondents showed higher prevalence (26.66%; OR=2.91; 95%CI OR=0.81-10.48) as compared to normal (11.11%; OR=Ref.) and overweight (10.29%; OR=0.92; 95%CI OR=0.35-2.43)

**Table 1. Seropositivity of brucellosis in pregnant women by SAT & ELISA.**

| *Brucella* Specie | (Positive/Total samples) (n/N) | Prevalence (%) (95%CI) |
|---|---|---|
| **Seropositivity by Serum Agglutination Test (SAT)** | | |
| *B. abortus* (*BA*) | 26/200 | 13.00 (8.82-18.36) |
| *B. melitensis* (*BM*) | 10/200 | 5.00 (2.56-8.83) |
| Both *BA* and *BM* | 07/200 | 3.50 (1.58-7.06) |
| **IgM captured ELISA** | | |
| *Brucella Spp.* | 24/200 | 12.00 (8.05-17.11) |

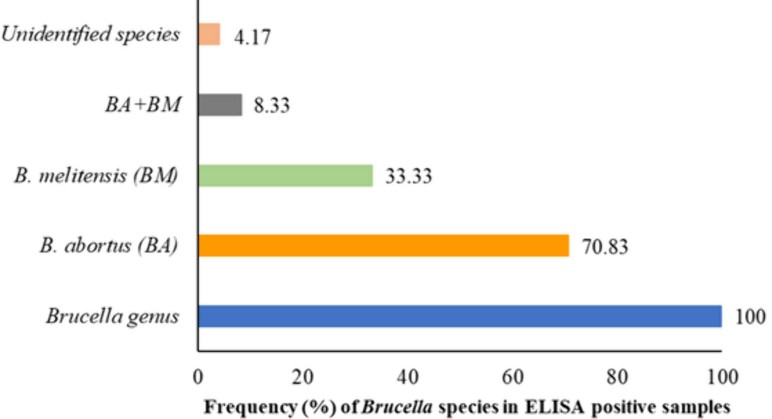

**Fig 3. PCR based distribution of *Brucella* species in ELISA-positive samples.**

respondents. Nutritional status of pregnant women was found to be a significant (p = 0.006; OR=3.325; 95%CI OR=1.309-8.219) risk factor and malnourished women showed higher prevalence rate of brucellosis (24.39%) as compared to well-nourished ones (8.81%). In this study, gestational age of pregnant women was also recorded as a significant risk factor (p = 0.017) with highest prevalence of 18.37% in 1st trimester (OR=Ref.) followed by 2nd trimester (7.23%; OR=0.35; 95%CI OR=0.13-0.92); however, no positive case was recorded in pregnant women having 3rd trimester (OR=0; 95%CI OR=0-∞). Result also showed that there was no association between number of pregnancies and prevalence of brucellosis in study population (p = 0.109).

The pregnant women with abortion history had significantly (p = 0.017; OR=3.215; 95%CI OR=1.11-8.581) higher prevalence rate (25.93%) as compared to those without any previous abortion history (9.83%). Similarly, previous history of premature births also had a significant (p = 0.021; OR=3.612; 95%CI OR=1.030-11.124) association with brucellosis. The educational status (p = 0.003; OR=4.506; 95%CI OR=1.611-16.424) and lack of awareness of respondents regarding brucellosis (p = 0.005; OR=0.237; 95%CI OR=0.065-0.664) were also significantly associated with prevalence of brucellosis in target population.

Results showed a non-significant difference (p = 0.384; OR=1.456; 95%CI OR=0.613-3.582) in prevalence rates of brucellosis in pregnant women of Lodhran (14%) and Vehari (10%). The residential location of pregnant women was found to be a significant risk factor (p = 0.014; OR=3.143; 95%CI OR=1.241-9.159) with higher prevalence rate (17.47%) in women residing in rural areas as compared to those residing in urban localities (6.19%). Source of drinking water remained a non-significant factor (p = 0.710; OR=1.173; 95%CI OR=0.490-2.961) in prevalence of brucellosis where 12.71% (n = 15/118) of the participants were using government supplied water and 10.98% (n = 09/82) were using motor pump as their drinking water source. Occupational status of respondents remained a non-significant factor (p = 0.424; OR=1.998; 95%CI OR=0.261-8.869). In study population, 133 respondents reported to have contact with *Brucella* susceptible animals (cattle, buffalo, sheep and goat) at their houses and/or workplaces with prevalence rate of 15.79% and this association was statistically significant (p = 0.020; OR=3.818; 95%CI OR=1.242-17.335). The consumption of raw/unboiled milk had a significant association with brucellosis in target population (p = 0.004; OR=4.107; 95%CI OR=1.392-11.294). A similar response was observed in pregnant women consuming semi and/or undercooked meat (p = 0.008; OR=3.612; 95%CI OR=1.237-9.771).

## Multivariate analysis

Multivariate analysis using BLRM with Nagelkerke's $R^2$ value of 0.583 was used to analyze the effect of predictors on results. Only significant risk factors of univariate analysis were included in BLRM. Results are shown in Table 4,

**Table 2. Association of host related risk factors with brucellosis – Univariate analysis.**

| Variable | Prevalence (n/N) (95%CI) | Odds Ratio (95%CI) | Pearson's | | Fisher's exact test |
| --- | --- | --- | --- | --- | --- |
| | | | Chi-square | P-value | P-value |
| **Age** | | | | | |
| ≤25 year | 17.33 (13/75) (9.965,27.725) | Ref | 3.296 | 0.192 | 0.228 |
| >25 but ≤40 | 9.18 (09/98) (4.685,16.539) | 0.48 (0.19,1.2) | | | |
| >40 but ≤50 | 7.41 (02/27) (1.330,23.259) | 0.38 (0.08,1.81) | | | |
| **BMI Category** | | | | | |
| Normal | 11.11 (13/117) (6.080,18.130) | Ref | 3.330 | 0.189 | 0.197 |
| Underweight | 26.66 (04/15) (9.657,53.530) | 2.91 (0.81,10.48) | | | |
| Overweight | 10.29 (07/68) (4.660,19.496) | 0.92 (0.35,2.43) | | | |
| **Nutritional Status** | | | | | |
| Malnourished | 24.39 (10/41) (12.573,40.024) | 3.325 (1.309,8.219) | 7.497 | 0.006 | 0.012 |
| Well-nourished | 8.81 (14/159) (5.118,14.262) | | | | |
| **Gestational Age** | | | | | |
| 1 Trimester | 18.37 (18/98) (11.374,27.346) | Ref | 8.143 | 0.017 | 0.016 |
| 2 Trimester | 7.23 (06/83) (3.189,14.703) | 0.35 (0.13,0.92) | | | |
| 3 Trimester | 0.0(00/19) (0.00,16.877) | 0.00 (0, infinity) | | | |
| **No of Pregnancy** | | | | | |
| No Birth | 17.14 (06/35) (7.732,32.479) | Ref | 6.058 | 0.109 | 0.110 |
| One Birth | 19.23 (10/52) (9.896,32.384) | 1.15 (0.38,3.52) | | | |
| Two Births | 7.32 (06/82) (3.232,14.888) | 0.38 (0.11,1.28) | | | |
| Three Births | 6.45 (02/31) (1.151,20.222) | 0.33 (0.06,1.79) | | | |
| **Abortion History** | | | | | |
| Yes | 25.93 (07/27) (11.804,46.165) | 3.215 (1.111,8.581) | 5.732 | 0.017 | 0.026 |
| No | 9.83 (17/173) (5.879,15.147) | | | | |
| **History of Premature Births** | | | | | |
| Yes | 29.41 (05/17) (12.374,54.428) | 3.612 (1.030,11.124) | 5.334 | 0.021 | 0.037 |
| No | 10.38 (19/183) (6.558,15.678) | | | | |

*(Continued)*

**Table 2.** (Continued)

| Variable | Prevalence (n/N) (95%CI) | Odds Ratio (95%CI) | Pearson's | | Fisher's exact test |
|---|---|---|---|---|---|
| | | | Chi-square | P-value | P-value |
| **Educational Status** | | | | | |
| Illiterate | 18.01 (20/111) (11.377,26.386) | 4.506 (1.611,16.424) | 8.555 | 0.003 | 0.004 |
| Literate | 4.49 (04/89) (1.548,10.859) | | | | |
| **Awareness regarding Brucellosis** | | | | | |
| Yes | 4.65 (04/86) (1.601,11.243) | 0.237 (0.065,0.664) | 7.716 | 0.005 | 0.007 |
| No | 17.54 (20/114) (11.078,25.686) | | | | |

n = positives samples, N = Total samples; Chi-square test; P-value <0.05 was considered significant.

where malnutrition (P = 0.037, OR=0.2431, 95%CI = 0.06450, 0.916), illiteracy (P = 0.049, OR=0.2168, 95%CI = 0.04742, 0.991), contact with *Brucella* susceptible animals (P = 0.009, OR=10.5142, 95%CI = 1.81622, 60.867), history of abortion (P = 0.012, OR=8.7308, 95%CI = 1.59961, 47.654), consumption of raw milk (P = 0.002, OR=37.1499, 95%CI = 3.94499, 349.840) and lack of awareness about brucellosis (P < 0.001, OR=0.0340, 95%CI = 0.00469, 0.246) were found to be significant risk factors. The model had an accuracy of 0.925, specificity of 0.977 and AUC of 0.932 with Receiver-operating characteristic curve (ROC) curve (Fig 4).

### Frequency of signs and symptoms in seropositive participants

During sampling, common signs and symptoms related to brucellosis were also recorded to analyze the frequency of different symptoms in *Brucella* positive participants. Results showed that fatigue with general weakness was most frequently (87.50%; n = 21) observed sign followed by myalgia (70.83%; n = 17), anorexia and weight loss (62.5%; n = 15 each), joint pain/ arthralgia (58.33%; n = 14), nausea and cough (29.16%; n = 7, each) and night sweats (20.83%; n = 5). (Table 5)

### Discussion

Brucellosis is endemic in ruminant population of Pakistan and due to its high zoonotic potential, its presence in humans is of utmost importance with respect to public health concern. Human brucellosis is directly associated with animal infection and in humans it can be avoided only by controlling the disease in animal population [7]. The risk of brucellosis in humans can increase many folds depending upon contact with animals and their products. In conventional livestock farming, women play a key role in different animal husbandry practices including nursing sick animals, assisting with births, cleaning up after livestock and dairy duties, which increases the likelihood/probability of their exposure to the bacteria and thus disease transmission [26].In Pakistan, most of the brucellosis investigations have been conducted on livestock [27–31]. Although few studies on human brucellosis have been reported in some parts of the country but studies particularly targeting the pregnant women with febrile condition which are a potentially high-risk group with reference to untoward consequences are scarce [21–23]. Accordingly, this study focused to assess the epidemiological status of brucellosis in pregnant women presented with febrile conditions at different reproductive healthcare facilities in the southern part of Punjab province of Pakistan. Results of our study showed a seropositivity rate of 13% and 5% for *B. abortus* and *B. melitensis* by SAT whereas IgM-ELISA showed an overall seroprevalence rate of 12% in target population. The PCR results confirmed the presence of *Brucella* genus in all ELISA-positive samples whereas PCR results reflected the presence of *B. abortus*

**Table 3. Association of socio-cultural factors with brucellosis – Univariate analysis.**

| Variable | Prevalence (n/N) (95%CI) | Odds Ratio (95%CI) | Pearson's | | Fisher's exact test |
|---|---|---|---|---|---|
| | | | Chi-square | P-value | P-value |
| **District Wise Distribution** | | | | | |
| Lodhran | 14.00 (14/100) (8.150,22.253) | 1.456 (0.613,3.582) | 0.758 | 0.384 | 0.515 |
| Vehari | 10.00 (10/100) (5.130,17.232) | | | | |
| **Residential Status** | | | | | |
| Rural | 17.48 (18/103) (10.819,26.006) | 3.143 (1.241,9.159) | 6.030 | 0.014 | 0.016 |
| Urban | 6.19 (06/97) (2.724,12.578) | | | | |
| **Source of Drinking Water** | | | | | |
| Filter (Govt Supply) | 12.71 (15/118) (7.403,20.098) | 1.173 (0.490,2.961) | 0.138 | 0.710 | 0.826 |
| Motor Pump | 10.98 (09/82) (5.633,19.775) | | | | |
| **Occupational Status** | | | | | |
| Employed | 20.00 (02/10) (3.671,55.559) | 1.998 (0.261,8.869) | 0.638 | 0.424 | 0.342 |
| Unemployed | 11.58 (22/190) (7.410,16.944) | | | | |
| **Contact with *Brucella* Susceptible Animals** | | | | | |
| Yes | 15.79 (21/133) (10.268,23.118) | 3.818 (1.242,17.335) | 5.399 | 0.020 | 0.021 |
| No | 4.48 (03/67) (1.223,12.203) | | | | |
| **Consumption of Raw Milk** | | | | | |
| Yes | 30.43 (07/23) (13.890,52.269) | 4.107 (1.392,11.294) | 8.364 | 0.004 | 0.009 |
| No | 9.60 (17/177) (5.725,14.802) | | | | |
| **Consumption of Semi/ Undercooked Meat** | | | | | |
| Yes | 28.00 (07/25) (12.762,47.938) | 3.612 (1.237,9.771) | 6.926 | 0.008 | 0.016 |
| No | 9.71 (17/175) (5.801,14.968) | | | | |

n = Positive samples, N = Total samples; Chi-square test; P-value <0.05 was considered significant.

and *B. melitensis* in 70.83% and 33.33% of ELISA positive samples. Contrary to our findings, Ali *et al.* [21] reported a lower prevalence rate of 5.8% in an observational study on pregnant women conducted at Benazir Bhutto Hospital located at Rawalpindi region of upper Punjab whereas Ali *et al.* [22] reported comparable prevalence of 16.36% among Pakistani women with spontaneous abortions at different hospitals of Lahore region of central Punjab. In Another study caried out at Haripur district of Khyber Pakhtunkhwa Pakistan reported a higher 23.63% seropositivity rate in women with spontaneous abortions [23]. Additionally, some previous studies also reported comparatively lower seroprevalence rates of 8% and 7% in apparently healthy women of Charsadda [32] and Kohat [33] regions of province Khyber Pakhtunkhwa-Pakistan, respectively. On the other hand, a study carried at Malakand region of Kyber Pakhtunkhwa also reported a

**Table 4. Association of significant risk factors of brucellosis in pregnant women – Multivariate Analysis.**

**Model Coefficients – Results**

| Predictor | Estimate | SE | Z | P-Value | Odds ratio | 95%CI | |
|---|---|---|---|---|---|---|---|
| | | | | | | Lower | Upper |
| Intercept | -1.616 | 1.031 | -1.56732 | 0.117 | 0.1987 | 0.02634 | 1.499 |
| **Nutritional status** | | | | | | | |
| Nourished -Malnourished | -1.414 | 0.677 | -2.08896 | 0.037 | 0.2431 | 0.06450 | 0.916 |
| **Residence** | | | | | | | |
| Urban - Rural | -0.730 | 0.737 | -0.99074 | 0.322 | 0.4817 | 0.11354 | 2.043 |
| **Gestational age (Trimesters)** | | | | | | | |
| 2 – 1 | -1.544 | 0.665 | -2.32298 | 0.020 | 0.2135 | 0.05802 | 0.786 |
| 3 – 1 | -18.738 | 2005.909 | -0.00934 | 0.993 | 7.28e-9 | 0.00000 | Infinity |
| **Education status** | | | | | | | |
| Literate - Illiterate | -1.529 | 0.775 | -1.97172 | 0.049 | 0.2168 | 0.04742 | 0.991 |
| **Contact with *Brucella* susceptible animals** | | | | | | | |
| Yes – No | 2.353 | 0.896 | 2.62605 | 0.009 | 10.5142 | 1.81622 | 60.867 |
| **History of abortions** | | | | | | | |
| Yes – No | 2.167 | 0.866 | 2.50248 | 0.012 | 8.7308 | 1.59961 | 47.654 |
| **History of premature deliveries** | | | | | | | |
| Yes – No | -0.320 | 1.030 | -0.31063 | 0.756 | 0.7262 | 0.09644 | 5.468 |
| **Consumption of semi/undercooked meat** | | | | | | | |
| Yes – No | 1.207 | 0.855 | 1.41182 | 0.158 | 3.3440 | 0.62583 | 17.868 |
| **Consumption of unboiled/raw milk** | | | | | | | |
| Yes – No | 3.615 | 1.144 | 3.15949 | 0.002 | 37.1499 | 3.94499 | 349.840 |
| **Awareness about brucellosis** | | | | | | | |
| Yes – No | -3.383 | 1.010 | -3.34848 | <.001 | 0.0340 | 0.00469 | 0.246 |

Estimates represent the log odds of "Results = Positive" vs. "Results = Negative".

higher brucellosis seropositivity rate of 27.47% in apparently healthy women [34]. Similarly, a study conducted in rural areas of Saudi Arabia also reported a lower prevalence rate of brucellosis (3.5%) in pregnant women [35] whereas in another hospital based study in Saudi Arabia reported an incident rate of 12.2% in pregnant women from 2005-2007 [36]. The varied findings of our study might be due to poor sanitation and hygiene practices, lack of awareness about the disease, and limited access to healthcare services in the study region. Moreover, the higher prevalence of brucellosis in this study is due to the selection of high-risk population as compared to the previous studies conducted in Pakistan [32–34]. Similar to our findings, both *B. abortus* and *B. melitensis* have been reported in pregnant women in different parts of the world [37]. Although brucellosis effects all age groups but highest prevalence of brucellosis was recorded among women with age less than 25 years followed by 25–40 years and the least prevalence noted in age 40–50 years. This can be explained by the fact that in rural areas women are married in younger age and it is the same age in which women are actively involved in house chores, looking after animals, and/ or involved in milking which increases their risk of contracting brucellosis. These results are in accordance to the previous studies conducted in Punjab, Pakistan [38]. Nutritional status of pregnant women was assessed to be a significant risk factor for brucellosis where malnourished pregnant women had the highest prevalence of brucellosis. Conversely, BMI category was found to be non-significant risk factor in the study population, however a previous study by Migisha *et. al* [39] reported a significant association between BMI and prevalence of brucellosis in general population The plausible reason for this difference might be that the pregnancy itself increases the weight of pregnant women which might alter the BMI category from underweight to normal or normal

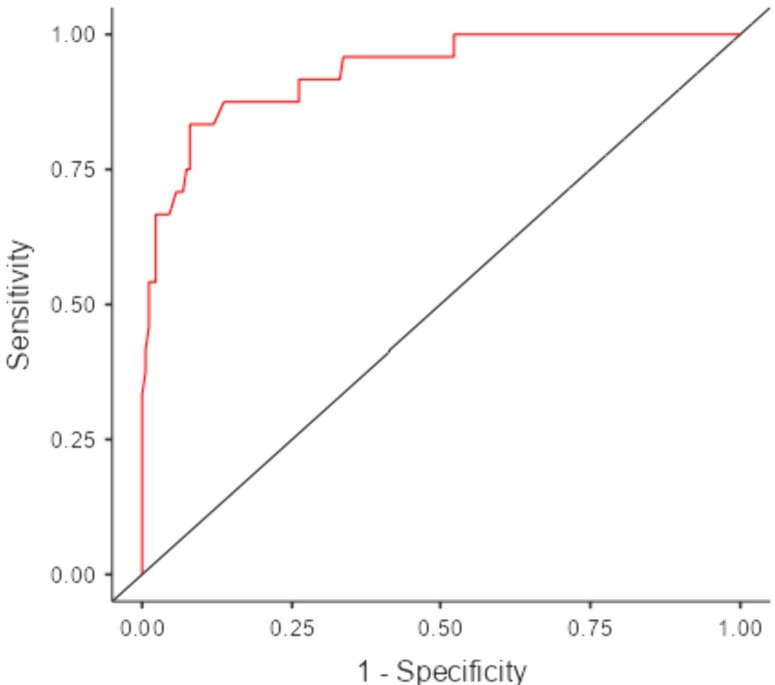

**Fig 4. Receiver-operating characteristic curve - Binomial Logistic Regression Model.**

**Table 5. Clinical signs and symptoms of brucellosis in pregnant women presented with febrile condition.**

| Signs and symptoms | *Brucella* Positive participants (n) | Participants with signs (n) | Signs/symptoms in B*rucella* positive participants (%) |
|---|---|---|---|
| Nausea | 24 | 7 | 29.16 |
| Cough | 24 | 7 | 29.16 |
| Anorexia & weight loss | 24 | 15 | 62.50 |
| Fatigue/Weakness | 24 | 21 | 87.50 |
| Night sweats | 24 | 5 | 20.83 |
| Joint pain/Arthralgia | 24 | 14 | 58.33 |
| Myalgia | 24 | 17 | 70.83 |

to overweight. Although non-significant, but the highest brucellosis prevalence was observed among women falling in underweight category. Gestational age, although a non-significant factor but absence of any positive case in participants from 3rd trimester pointed towards brucellosis causing miscarriages, preterm labors, premature births in early pregnancy and these results are in conformity with research carried out in Saudi Arabia on 92 pregnant women with acute brucellosis [37]. As per our study, number of previous pregnancies have no relationship with brucellosis and these results are in accordance with the study carried out in Diarbakir, Turkey to evaluate effects of brucellosis in pregnancy [40]. History of abortions in pregnant women with the highest prevalence rate (25.93%) remained a significant predisposing factor towards brucellosis in this study. Similar findings have been reported in similar previous studies [21,41]. Pregnant women with history of premature births have also shown highest prevalence rate (29.41%). These results mutually imply that brucellosis can become chronic and can cause abortions and premature births in future pregnancies as well. These findings also point towards multiple reinfections if not treated.

Lack of education and lack of awareness about brucellosis are main causes of its zoonotic transmission cycle [42]. In our study, illiterate women had the highest prevalence (18.01%) whereas people with knowledge regarding brucellosis had lower prevalence rate of disease (4.65%). This shows that appropriate awareness regarding brucellosis can prevent the risk of its spread [43].

In the current study, an apparently higher prevalence rate of brucellosis was observed in Lodhran as compared to Vehari and this difference might be attributed to the higher rural population of Lodhran (84.37%) as compared to Vehari (82.56%) [24]. Rural residential status remained a significant risk factor for transmission of infection, and similar findings have been reported by a study conducted in Peshawar, Pakistan [44]. This difference might be due to the fact that rural communities have more frequent contact with *Brucella* susceptible animals and comparatively lower access to personal protective equipment as compared to urban population. Additionally, lower literacy status and poor hygiene practices in rural areas might also contribute to the higher prevalence of brucellosis in rural populations. Humans are accidental hosts of *Brucella* and disease in humans is mainly linked to *Brucella* affected animal sources which serve as reservoir hosts of this bacteria posing a risk of zoonotic transmission of infection [45]. Thus, animal contact, consumption of raw milk/dairy products and consumption/ processing of undercooked meat are major risk factors for transmission of brucellosis [46,47]. Corresponding results have been obtained in our study where animal contact remained a predisposing factor in transmission with significantly higher (15.79%) prevalence rate of brucellosis in target population. Consumption of raw milk and its products were found to be significant risk factors towards disease spread with higher prevalence (30.43%) among women with habit of consuming raw milk or dairy products including yogurt, cheese, cream, and ghee made from raw milk. Similarly, statistical analysis revealed a significant association between consumption of semi/ under cooked meat and prevalence of brucellosis in the target population. Additionally, correlation of drinking water source with prevalence of brucellosis was also analyzed in this study but the results remained non-significant towards disease transmission due to non-shared sources of water among humans and animals. Urbanicity, history of premature births, consumption of undercooked or semi-cooked meat and gestational age are non-significant risk factors in disease transmission by binomial logistic regression while calculated as significant risk factors by chi-square test. These results should be considered together to gain a comprehensive understanding of the factors associated with disease transmission.

## Conclusions and recommendations

Brucellosis is endemic in pregnant women presented with with febrile illness at different healthcare facilities of Lodhran and Vehari districts of southern Punjab-Pakistan. Malnutrition, poor educational status, contact with *Brucella* susceptible animals, abortion history, consumption of raw milk and lack of awareness about brucellosis have significant association with brucellosis among pregnant women with febrile condition. The women dealing with animals may be at high risk to contact the zoonotic infections including brucellosis. Therefore, it is recommended that disease control/prevention measures should be developed and operationalized at regional and national levels. Among various measures, public awareness campaigns on zoonotic transmission of brucellosis appears to be the most promising and readily doable risk mitigation measure that should be intensified in rural/high-risk areas. Further, controlling the disease in susceptible animals by immunization may also help in lowering the disease burden in human beings which are dead end hosts of *Brucella* species. It is also suggested to develop a consortium of human and veterinary physicians to tackle this important veterinary public health issue in one health perspective for its better understanding and control at animal-human interface.

## Study limitations

Convenient sampling was used for collection of data due to specificity of target group. To overcome bias in data, healthcare facilities with maximum influx of pregnant women were targeted which led to diversified population and make this study reliable and generalizable.

## Supporting information

**S1 Data.  Raw data used to compute the seroprevalence of brucellosis and its association with different risk factors in study population.** All data presented in supplementary information is self-explanatory, no legends required. (XLSX)

## Acknowledgments

The authors highly acknowledge the health professionals of Vehari and Lodhran for their assistance in recruiting the study participants from target population for collection of blood samples and descriptive epidemiological data.

## Author contributions

**Conceptualization:** Mian Muhammad Awais, Masood Akhtar.

**Data curation:** Syed Muhammad Tauseef Shafqat, Mian Muhammad Awais, Abdul Sammad Ali Khan Shirwany, Imran Rasool.

**Formal analysis:** Syed Muhammad Tauseef Shafqat, Mian Muhammad Awais, Ulas Acaroz, Abdul Sammad Ali Khan Shirwany, Imran Rasool.

**Funding acquisition:** Mian Muhammad Awais, Masood Akhtar.

**Investigation:** Mian Muhammad Awais, Masood Akhtar, Naseer Ali Shah.

**Methodology:** Syed Muhammad Tauseef Shafqat, Mian Muhammad Awais, Ulas Acaroz, Masood Akhtar, Abdul Sammad Ali Khan Shirwany, Imran Rasool, Naseer Ali Shah.

**Project administration:** Mian Muhammad Awais, Masood Akhtar.

**Resources:** Mian Muhammad Awais, Ulas Acaroz, Imran Rasool.

**Software:** Syed Muhammad Tauseef Shafqat, Mian Muhammad Awais.

**Supervision:** Mian Muhammad Awais, Naseer Ali Shah.

**Validation:** Mian Muhammad Awais, Ulas Acaroz, Masood Akhtar, Abdul Sammad Ali Khan Shirwany, Imran Rasool, Naseer Ali Shah.

**Visualization:** Mian Muhammad Awais, Abdul Sammad Ali Khan Shirwany.

**Writing – original draft:** Syed Muhammad Tauseef Shafqat, Mian Muhammad Awais, Abdul Sammad Ali Khan Shirwany.

**Writing – review & editing:** Syed Muhammad Tauseef Shafqat, Mian Muhammad Awais, Masood Akhtar, Naseer Ali Shah.

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
