## [Editor Report · Decision Letter 0]

26 Aug 2025

PNTD-D-25-01396Exploring the status of brucellosis in pregnant women presented with febrile illness at different healthcare facilities of Vehari and Lodhran zones of PakistanPLOS Neglected Tropical DiseasesDear Dr. Mian Muhammad Awais, Thank you for submitting your manuscript to PLOS Neglected Tropical Diseases. After careful consideration, we feel that it has merit but does not fully meet PLOS Neglected Tropical Diseases's publication criteria as it currently stands. Therefore, we invite you to submit a revised version of the manuscript that addresses the points raised during the review process. Please submit your revised manuscript within 30 August 2025. If you will need more time than this to complete your revisions, please reply to this message or contact the journal office at plosntds@plos.org. Please include the following items when submitting your revised manuscript: * A rebuttal letter that responds to each point raised by the editor and reviewer(s). You should upload this letter as a separate file labeled 'Response to Reviewers '. This file does not need to include responses to any formatting updates and technical items listed in the 'Journal Requirements' section below. * A marked-up copy of your manuscript that highlights changes made to the original version. You should upload this as a separate file labeled 'Revised Manuscript with Track Changes '. * An unmarked version of your revised paper without tracked changes. You should upload this as a separate file labeled 'Manuscript '. If you would like to make changes to your financial disclosure, competing interests statement, or data availability statement, please make these updates within the submission form at the time of resubmission. Guidelines for resubmitting your figure files are available below the reviewer comments at the end of this letter. We look forward to receiving your revised manuscript. Kind regards, KHM Nazmul Hussain Nazir, DVM, PhDGuest EditorPLOS Neglected Tropical Diseases Ana LTO NascimentoSection EditorPLOS Neglected Tropical Diseases

Shaden Kamhawi

co-Editor-in-Chief

Paul Brindley

co-Editor-in-Chief

**Additional Editor Comments:** The author improved the article significantly. The article has novelty- no doubt. Brucellosis is considered a neglected disease. The approach and findings may add some value, particularly for a a developing country like Pakistan.

However, before acceptance, please send the article back to the author for the minor corrections again.

Corrections:

Lines 67 to 70: Please divide the long sentence into 2-3 separate sentences.

Line 73: A reference is needed.

Line 77: Quantify loss with a reference.

Line 79: Need recent reference.

Line 93: The reference is not appropriate.

Line 97: The reference is not accessible.

Line 103: Several references have been cited, but not all references report the same data.

Line 144: Add exclusion criteria.

Line 300: Need a reference. **Journal Requirements:**

1) Please upload all main figures as separate Figure files in .tif or .eps format. For more information about how to convert and format your figure files please see our guidelines: 

2) We have noticed that you have uploaded Supporting Information files, but you have not included a list of legends. Please add a full list of legends for your Supporting Information files after the references list.

3) Please ensure that the funders and grant numbers match between the Financial Disclosure field and the Funding Information tab in your submission form. Note that the funders must be provided in the same order in both places as well.

**Reviewers' comments:** **Figure resubmission:** While revising your submission, we strongly recommend that you use PLOS’s NAAS tool (https://ngplosjournals.pagemajik.ai/artanalysis) to test your figure files. NAAS can convert your figure files to the TIFF file type and meet basic requirements (such as print size, resolution), or provide you with a report on issues that do not meet our requirements and that NAAS cannot fix.

After uploading your figures to PLOS’s NAAS tool - https://ngplosjournals.pagemajik.ai/artanalysis, NAAS will process the files provided and display the results in the "Uploaded Files" section of the page as the processing is complete. If the uploaded figures meet our requirements (or NAAS is able to fix the files to meet our requirements), the figure will be marked as "fixed" above. If NAAS is unable to fix the files, a red "failed" label will appear above. When NAAS has confirmed that the figure files meet our requirements, please download the file via the download option, and include these NAAS processed figure files when submitting your revised manuscript. **Reproducibility:** To enhance the reproducibility of your results, we recommend that authors of applicable studies deposit laboratory protocols in protocols.io, where a protocol can be assigned its own identifier (DOI) such that it can be cited independently in the future. Additionally, PLOS ONE offers an option to publish peer-reviewed clinical study protocols. Read more information on sharing protocols at https://plos.org/protocols?utm_medium=editorial-email&utm_source=authorletters&utm_campaign=protocols

---

## [Editor Report · Decision Letter 1]

3 Sep 2025

Dear Dr. Awais,

We are pleased to inform you that your manuscript 'Exploring the status of brucellosis in pregnant women presented with febrile illness at different healthcare facilities of Vehari and Lodhran zones of Pakistan' has been provisionally accepted for publication in PLOS Neglected Tropical Diseases.

Best regards,

KHM Nazmul Hussain Nazir, DVM, PhD

Guest Editor

Ana LTO Nascimento

Section Editor

Shaden Kamhawi

co-Editor-in-Chief

Paul Brindley

co-Editor-in-Chief

---

## [Editor Report · Acceptance letter]

Dear Dr Awais,

We are delighted to inform you that your manuscript, "Exploring the status of brucellosis in pregnant women presented with febrile illness at different healthcare facilities of Vehari and Lodhran zones of Pakistan," has been formally accepted for publication in PLOS Neglected Tropical Diseases.

Best regards,

Shaden Kamhawi

co-Editor-in-Chief

Paul Brindley

co-Editor-in-Chief
